# The Levels of Circulating MicroRNAs at 6-Hour Cardiac Arrest Can Predict 6-Month Poor Neurological Outcome

**DOI:** 10.3390/diagnostics11101905

**Published:** 2021-10-15

**Authors:** Sang Hoon Oh, Ho-Shik Kim, Kyu Nam Park, Sanghee Ji, Ji-Young Park, Seung Pill Choi, Jee Yong Lim, Han Joon Kim

**Affiliations:** 1Department of Emergency Medicine, Seoul St. Mary’s Hospital, College of Medicine, The Catholic University of Korea, Seoul 06591, Korea; ohmytweety@catholic.ac.kr (S.H.O.); emsky@catholic.ac.kr (K.N.P.); ny1117@catholic.ac.kr (J.Y.L.); 2Department of Biochemistry, College of Medicine, The Catholic University of Korea, Seoul 06591, Korea; hoshik2013@gmail.com (H.-S.K.); tkdgml675@naver.com (S.J.); jpweb@catholic.ac.kr (J.-Y.P.); 3Department of Emergency Medicine, Eunpyeong St. Mary’s Hospital, College of Medicine, The Catholic University of Korea, Seoul 03312, Korea; emvic98@catholic.ac.kr

**Keywords:** heart arrest, induced hypothermia, prognostication, biomarker, microRNA

## Abstract

Early prognostication in cardiac arrest survivors is challenging for physicians. Unlike other prognostic modalities, biomarkers are easily accessible and provide an objective assessment method. We hypothesized that in cardiac arrest patients with targeted temperature management (TTM), early circulating microRNA (miRNA) levels are associated with the 6-month neurological outcome. In the discovery phase, we identified candidate miRNAs associated with cardiac arrest patients who underwent TTM by comparing circulating expression levels in patients and healthy controls. Next, using a larger cohort, we validated the prognostic values of the identified early miRNAs by measuring the serum levels of miRNAs, neuron-specific enolase (NSE), and S100 calcium-binding protein B (S100B) 6 h after cardiac arrest. The validation cohort consisted of 54 patients with TTM. The areas under the curve (AUCs) for poor outcome were 0.85 (95% CI (confidence interval), 0.72–0.93), 0.82 (95% CI, 0.70–0.91), 0.78 (95% CI, 0.64–0.88), and 0.77 (95% CI, 0.63–0.87) for miR-6511b-5p, -125b-1-3p, -122-5p, and -124-3p, respectively. When the cut-off was based on miRNA levels predicting poor outcome with 100% specificity, sensitivities were 67.7% (95% CI, 49.5–82.6), 50.0% (95% CI, 32.4–67.7), 35.3% (95% CI, 19.7–53.5), and 26.5% (95% CI, 12.9–44.4) for the above miRNAs, respectively. The models combining early miRNAs with protein biomarkers demonstrated superior prognostic performance to those of protein biomarkers.

## 1. Introduction

Since positive results of two landmark studies were reported in 2002 [1,2], targeted temperature management (TTM) has been accepted as the intervention to improve outcomes for patients resuscitated from cardiac arrest (CA). Currently, the recommendations for TTM from the European Resuscitation Council include maintaining a target temperature for all comatose cardiac arrest survivors at a constant value between 32 °C and 36 °C for at least 24 h and avoiding fever for at least 72 h after the return of spontaneous circulation (ROSC) in patients who remain in a coma [3]. Most deaths in patients with TTM are caused by irreversible ischemic brain injury [4]. However, approximately one-third of deaths are due to non-neurological causes, which usually occur in the first 3 days after the return of spontaneous circulation (ROSC) [5]. Therefore, early detection of whether they already have hypoxic brain injury or not is important for allowing further advanced treatment decisions and guiding post-cardiac arrest management strategies. However, early prognosis is challenging for physicians. Among commonly used prognostic tools, neurological examination is not always accurate due to the use of medications during TTM [6]. The detection of ischemic changes on neuroimaging tests can be delayed [7], and the results of electrophysiologic studies such as electroencephalography (EEG) and somatosensory evoked potential (SSEP) are inconsistent during the first days after CA [8,9]. Furthermore, interrater variability is frequently demonstrated with these tools [10,11]. While EEG findings are described with a standardized terminology, the skill of EEG interpretation is learned primarily in a master–apprentice format [12]. In a recent study, full-length EEGs from 103 comatose cardiac arrest patients were interpreted by four EEG specialists who were blinded for patient outcome; there was moderate interrater agreement for malignant EEG patterns [10]. The main source of disagreement in SSEP interpretation was related to the noise levels [13]. When SSEPs were interpreted independently by experienced clinical neurophysiologists, moderate interobserver agreement was found [11,13]. In contrast, serum biomarkers are unlikely to be affected by medications and are easy to assess without interrater variability.

Neuron-specific enolase (NSE) and S100 calcium-binding protein B (S100B) are the only biomarkers that have been used for the prediction of outcomes after CA [14,15]. However, current guidelines do not recommend a biomarker to predict a poor outcome in the early phase. MicroRNAs (miRNAs) are noncoding RNA molecules composed of 19–24 nucleotides that regulate gene expression by inhibiting or inactivating target messenger RNAs [16]. Recently, results of advanced molecular biology techniques have revealed that miRNAs can be used as candidate biomarkers in CA patients. After CA, neuronal miRNAs cross the disrupted blood–brain barrier (BBB) and can be measured in the plasma [17,18,19]; recent studies have reported the prognostic utility of these miRNAs [20,21,22]. While previous studies have assessed the prognostic value of circulating brain-enriched miRNAs at 48 h after CA, cardiac-enriched miRNAs are released early after cardiac injury [23]; Wander et al. demonstrated that circulating miRNAs during CA are related to resuscitation outcome [24].

Therefore, in the present study, we hypothesized that in CA patients with TTM, circulating miRNA expression at 6 h after ROSC is associated with 6-month neurological outcomes. We performed miRNA sequencing technology in a small subset of samples and then validated the differentially expressed miRNAs in a larger cohort using quantitative real-time polymerase chain reaction (qRT-PCR).

## 2. Materials and Methods

### 2.1. Study Protocol and Subjects

This prospective observational study was conducted in a single tertiary hospital in Seoul, South Korea. The study consisted of two phases: a discovery phase and a validation phase. The discovery phase aimed to identify candidate miRNAs for neurological prediction after CA. From June to July 2016, a convenience sample of 11 adult (>18 years of age) patients with TTM at 6 h after ROSC and 4 healthy controls were examined. We identified miRNAs with significantly different serum expression between the two groups. For the validation phase, we measured the expression of candidate miRNAs and the serum levels of NSE and S100B at 6 h after ROSC in consecutive TTM-treated patients from September 2016 to July 2018 (*n* = 54). During the study period, all comatose patients with ROSC were considered eligible for TTM at 33 °C or 36 °C for 24 h. The exclusion criteria included unavailable serum samples. This study was approved by our Institutional Review Board, and written informed consent was obtained from each patient’s next of kin.

### 2.2. Small RNA Sequencing

Small RNA sequencing libraries were constructed using the NEXTflex Small RNA sample preparation protocol with 70 ng of total RNA as input. The adapters were directly ligated to the total RNA as follows. For the first step, NEXTflex 3’ 4N adenylated adapter (5′ rApp/NNNNTGGAATTCTCGGGTGCCAAGG/3ddC/) and NEXTflex 5’ 4N adapter (5’ GUUCAGAGUUCUACAGUCCGACGAUCNNNN) were ligated to each end of the total RNA samples. The 5′ and 3′ NEXTflex adapter-ligated products were reverse transcribed by M-MuLV reverse transcriptase in the presence of RNA RT primer (5′ GCCTTGGCACCCGAGAATTCCA) to produce single-stranded complementary DNA (cDNA). The cDNA was then PCR amplified using a universal primer (5’ AATGATACGGCGACCACCGAGATCTACACGTTCAGAGTTCTACAGTCCGA) and a primer containing barcode sequences for 18 cycles. The PCR cycles consisted of 20 s at 95 °C, 30 s at 60 °C, and 15 s at 72 °C. The amplified cDNA was separated on a 6% TBE gel (Invitrogen, Waltham, MA, USA), and the 140–160 bp bands were excised. The cDNA was then eluted from the gel and concentrated by ethanol precipitation. The quality and size distribution of the adapter-ligated RNAs and amplified libraries were confirmed by electrophoresis on Agilent Bioanalyzer High Sensitivity DNA microfluidic chips (Agilent, Santa Clara, CA, USA). Libraries were quantified using the KAPA Library Quantification Kit (KK4824, Kapa Biosystems, Wilmington, MA, USA).

Subsequently, the library was sequenced using an Illumina HiSeq2500 that was set to rapid run mode. Cluster generation, followed by 2 × 100 cycle sequencing reads separated by paired-end turnaround, was performed on the instrument. Image analysis was performed using HiSeq Control Software version 1.8.4. The high-quality reads were then mapped onto the human reference genome (ENSEMBL release 72) using bowtie with the following parameters: only one mismatch (−n 1), max 80 sum of mismatch quals across alignment, 30 seed length, and the suppression of the reads showing >5 alignments [25,26]. The miRNA reads were counted using HTSeq with the ‘intersection-nonempty’ mode based on miRBase release 20 [27,28]. Based on the read counts for each miRNA, EdgeR was applied to analyze the differential expression between regions [29]. Differentially expressed miRNAs were identified with a significant *p*-value of < 0.05.

### 2.3. qRT-PCR

Small RNA was extracted from plasma using the miRNeasy Serum/Plasma Kit (Qiagen, Hilden, Germany). Small RNA from three 200 µL-plasma aliquots was extracted according to the manufacturer’s instructions, and finally, RNA was eluted with RNase-free water. The purity and quantity of extracted RNA were measured using a NanoDrop ND-2000 (Thermo Fisher Scientific, Wilmington, DE, USA). Since appropriate endogenous reference genes are absent in plasma, UniSp6 RNA was added to the extracted RNA samples before the cDNA synthesis reaction. First-strand cDNA synthesis and qPCR were performed using a miRCURY LNA miRNA PCR Assay kit (Qiagen, Hilden, Germany). After first-strand cDNA synthesis was completed, the cDNA was amplified by miRNA-specific and LNA-enhanced primers using an ABI 7300 Real-Time PCR System (Applied Biosystems, Carlsbad, CA, USA). Each miRNA level was normalized to the UniSp6 RNA level in the same sample, and the relative changes across samples were expressed relative to the spike-in UniSp6 using the ΔΔCt method [30].

### 2.4. Laboratory Measurement

NSE and S100B measurements were performed with Roche Elecsys reagents specific to NSE and S100B (Roche Diagnostics, Mannheim, Germany) in serum at 6 h after ROSC. If the serum showed significant hemolysis, the results were discarded. The serum levels of troponin T, N-terminal pro-brain natriuretic peptide, and total bilirubin were measured at admission.

### 2.5. Outcome Measurement

The primary endpoint was a 6-month poor neurological outcome defined as a cerebral performance category (CPC) score of 3–5. A CPC score of 1–2 indicated a good outcome. Additionally, to evaluate the hemodynamic state, whether the patient had a shock on admission and needed vasopressor support at 6 h was assessed.

### 2.6. Statistical Analysis 

All data are summarized and displayed as the number (percentage) of patients in each group for categorical variables and as the median with interquartile range (IQR) for continuous variables. Comparisons of categorical variables between groups were made using Fisher’s exact test. After testing for normal distribution, continuous variables were compared using Mann–Whitney U tests. The ability of miRNAs and NSE or S100B to predict poor outcomes was assessed based on their sensitivity and specificity using an exact binomial 95% confidence interval (CI) and receiver operating characteristic (ROC) curve analysis. Pairwise area under the curve (AUC) comparisons were also performed between two predictors using the nonparametric approach [31]. We also created combined models using several logistic regression models and compared the AUCs of single biomarkers. The Pearson correlation coefficients between biomarkers were calculated (low, 0.10–0.29; moderate, 0.30–0.69; strong, 0.70–1.00). All analyses were performed using SPSS 24.0 software (IBM, SPSS Inc., Chicago, IL, USA). A value of *p* < 0.05 was considered significant for all analyses.

## 3. Results

### 3.1. Discovery Phase

We first performed small RNA sequencing to profile the expression of miRNAs in 4 healthy controls and 11 CA patients (good outcome, 5; poor outcome, 6). Baseline characteristics were not significantly different between the CA and control groups (Appendix A). A median of 413 (IQR, 371–542) known miRNAs and 179 (IQR, 130–254) novel miRNAs were identified in the 15 samples (Appendix A).

### 3.2. Validation Phase

From the differentially expressed miRNAs identified by small RNA sequencing, we selected 10 miRNAs (miR-6511b-5p, -125b-1-3p, -122-5p, -124-3p, -18b-3p, -511-5p, -519a-3p, -3180-3p, -24-2-5p, and -590-3p) and validated these miRNAs using qRT-PCR. The validation cohort consisted of 54 CA patients treated with TTM, of whom 20 patients had a good outcome and 34 had a poor outcome. The patients’ characteristics are summarized in Table 1. There was no difference between the outcome groups with regard to age, sex, comorbidities, or bystander cardiopulmonary resuscitation and witnessed arrest. Patients with poor outcomes had a higher rate of non-cardiac etiology arrest, non-shockable initial rhythm, and longer time to ROSC than those with good outcomes (all *p* < 0.05).

The circulating levels of the 10 candidate miRNAs at 6 h after ROSC were measured using qRT-PCR. All miRNAs showed significant differential expression between the outcome groups (Appendix A). Among these, four miRNAs (miR-6511b-5p, -125b-1-3p, -122-5p, and -124-3p) for which the expression difference between the groups was most significant were selected for additional statistical analysis.

To explore the function of these miRNAs, we analyzed gene ontologies using the PANTHER resource (http://pantherdb.org/) (Appendix A) [32].

The circulating levels of the miRNAs and the serum levels of NSE and S100B according to the outcome group are shown in Figure 1 and Figure 2, respectively. ROC analysis of miR-6511b-5p, -125b-1-3p, -122-5p, and -124-3p showed AUCs of 0.85 (95% CI, 0.72–0.93), 0.82 (95% CI, 0.70–0.91), 0.78 (95% CI, 0.64–0.88), and 0.77 (95% CI, 0.63–0.87), respectively (Figure 3). When the cutoff was based on miRNA levels predicting poor outcome with 100% specificity, the sensitivities for the above miRNAs were 67.7% (95% CI, 49.5–82.6), 50.0% (95% CI, 32.4–67.7), 35.3% (95% CI, 19.7–53.5), and 26.5% (95% CI, 12.9–44.4), respectively. In addition, the AUCs of NSE and S100B measured at 6 h were 0.72 (95% CI, 0.58–0.86) and 0.85 (95% CI, 0.75–0.96), respectively, and were not significantly different from the AUCs of the four miRNAs (all *p* > 0.05). A cutoff of >63.52 ng mL^−1^ for NSE and >0.379 ng mL^−1^ for S100B yielded 100% specificity and 26.5% and 32.6% sensitivities for a poor outcome, respectively.

According to the hemodynamic state, differences in the expression levels of miR-6511b-5p, -125b-1-3p, and -122-5p were also observed (Appendix A).

### 3.3. Correlations between Biomarkers

The levels of miR-6511b-5p, -125b-1-3p, and -124-3p moderately correlated with the levels of NSE (r = 0.502, r = 0.479, and r = 0.456, respectively) and S100B (r = 0.344, r = 0.428, and r = 0.432, respectively) (Figure 4). However, there was no correlation between miR-122-5p and NSE or S100B (r = 0.076, *p* = 0.584 and r = 0201, *p* = 0.144, respectively). Regarding inter-miRNA expression, the correlation between miR-6511b-5p and miR-125b-1-3p was the strongest (r = 0.920, *p* < 0.001), and there was no correlation between miR-122-5p and miR-124-3p (r = 0.070, *p* = 0.616). The correlation coefficients between miRNAs and other laboratory values are presented in Appendix A.

### 3.4. Prognostic Performance of miRNAs Combined with Protein Biomarkers

The AUCs of the different logistic regression models with combinations of various miRNAs added to either or both NSE and S100B were calculated (Table 2). The AUCs to predict poor outcomes at 6 months increased when various miRNAs were added to NSE or S100B. In particular, adding miR-6511b-5p or -125b-1-3p significantly improved the AUC of NSE (0.87 (95% CI, 0.78–96) vs. 0.72 (95% CI, 0.58–0.83), *p* = 0.029 and 0.86 (95% CI, 0.76–0.96) vs. 0.72 (95% CI, 0.58–0.83), *p* = 0.027, respectively). However, miR-122-5p and -124-3p did not. This trend was similar to the results of combining miRNAs with S100B or both NSE and S100B, although the differences were not significant (all *p* > 0.05).

## 4. Discussion

Our results indicated that the early expression of several miRNAs in the serum at 6 h after ROSC can predict 6-month neurological outcome in CA patients treated with TTM. In particular, miR-6511b-5p and -125b-1-3p predicted poor outcomes, with moderate sensitivity. The models combining early miRNAs with early NSE or S100B demonstrated superior performance to those of either NSE or S100B alone.

As a biomarker of ischemic stroke, miR-124-3p, a brain-enriched miRNA, has been well characterized previously [33,34]. In CA, Gilje et al. presented that circulating levels of miR-124-3p were increased in patients with poor outcome compared to those with good outcome at 24 h and 48 h after CA [21]. Further large-scale studies conducted in 579 cohorts confirmed the prognostic value of miR-124-3p at 48 h [35]. We identified miR-124-3p as a possible predictor even at 6 h after ROSC. This result is supported by several other studies [17,19,35]. Stefanizzi et al. reported that the circulating levels of brain-enriched miRNAs at 48 h after CA correlated with NSE levels [36]. Interestingly, according to our results, the circulating levels of miR-124-3p were also correlated with the serum levels of NSE and S100B at 6 h after CA.

In one pioneering study, miR-122 was overexpressed in the serum of poor outcome patients at 48 h after CA [20]; however, the prognostic values of miR-122 are inconsistent among different studies. Gilje et al. were unable to confirm that miR-122 has prognostic value after CA and found a decrease in the plasma levels of miR-122 in both outcome groups [21]. Another large-scale study reported that patients with low levels of miR-122-5p were at high risk for poor outcomes [37]. miR-122 is generally regarded as a liver-specific miRNA and is involved in lipid metabolism [38]. Our results indicated that, although increased miR-122-5p expression predicted poor outcome, it was not correlated with NSE, S100B, or miR-124-3p expression. Several possibilities might account for these variable results. First, the characteristics of the included cohorts between studies were different. Our ratios of non-shockable rhythm arrest and non-cardiac etiology arrest were generally higher than those in other published reports with opposite results. Post-CA syndrome is a heterogeneous entity that involves multiple organs as well as the brain and causes death through a variety of mechanisms. In our study, miR-122-5p expression was associated with bilirubin levels and the presence of shock, which induces multiple organ damage, such as damage to the liver. However, it was observed that these associations were not specific to miR-122-5p. Second, the temperature at the timing of miRNA sampling could impact miRNA expression. Hypothermia regulates miRNA expression by enhancing the processing of pre-miRNAs by Dicer [39,40,41]. In the porcine cardiogenic shock model, mildly induced hypothermia decreased the plasma levels of miR-122 [42]. Therefore, miRNA expression at the early phase of hypothermia could be different from that of other studies.

The miRNAs identified as prognostic candidate biomarkers in our study deserve further acknowledgement. The prognostic values of miR-6511b-5p and -125b-1-3p were superior to those of well-known miRNA and protein biomarkers, and their high expressions had 100% specificity with moderate sensitivity for poor outcomes. miR-125b has been demonstrated to negatively regulate NR2A-containing N-methyl-D-aspartate receptor [43], while activation of this glutamate receptor exerts neuroprotective effects and promotes neuronal survival against excitotoxicity-mediated neuronal damage [44]. Meanwhile, miR-6511b-5p has been poorly characterized. Since their discovery in 2001, miRNAs have become potential biomarkers of neurological and cardiovascular disease. However, they have rarely been studied in CA. Further studies are warranted to confirm that miR-6511b-5p has good prognostic performance for these patients. 

Early detection and stratification of brain injury can help clinicians optimize the dose of in-hospital treatment [45]. When combined with protein biomarkers, the miRNAs showed higher prognostic value. The combination of NSE, S100B, and miR-6511b-5p has the best AUC value, and interestingly, models adding a miRNA to NSE are slightly better than the S100B model. This might be due to the kinetics of S100b, which is an “earlier” biomarker than NSE, with a tendency to decrease over time, compared to NSE [46,47].

Strengths of this study include the various protein biomarkers available and systematic approach to discover various miRNA candidates. Because patients with non-neurological death could be categorized into the poor outcome group, we used NSE and S100B as surrogate markers of brain injury. We also performed miRNA sequencing and then validated the differentially expressed miRNAs. Through this approach, we identified several miRNAs for early prediction.

Our study should be interpreted in the context of the following limitations. First, this study included a relatively small number of patients. Therefore, we could not adjust variables to evaluate the mechanisms underlying these observed differences in miRNA expression. Second, although the aim of the present study was to investigate miRNA candidates as early biomarkers, we did not conduct serial measurements of miRNAs. The kinetics of release of brain-derived miRNAs after CA could not be accurately characterized. Finally, although miRNAs have been reported to have some potential to attenuate ischemic brain injury [48], we only focused on the prognostic value of miRNA with increased expression as a poor outcome predictor. Thus, further studies using a larger sample size are necessary to confirm our findings and determine the precise reasons for the observed differences in miRNA expression.

## 5. Conclusions

In this small sample study, we identified several miRNAs for the early prediction of 6-month neurological outcomes in TTM-treated CA patients. In particular, high expressions of miR-6511b-5p and -125b-1-3p at 6 h after ROSC predicted poor outcomes, with moderate sensitivities, and may complement other biomarkers. Our results need to be validated in larger cohorts.

## Figures and Tables

**Figure 1 diagnostics-11-01905-f001:**
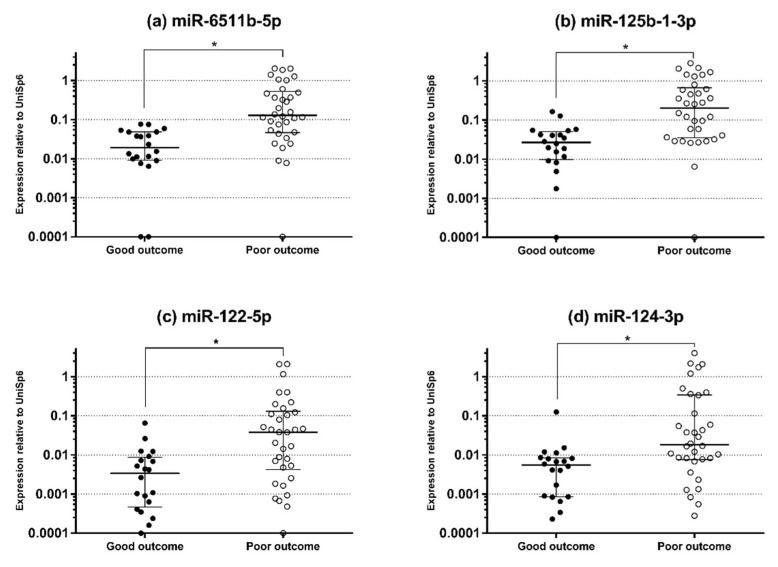
Serum expression levels of various miRNAs at 6 h after return of spontaneous circulation in patients with good and poor neurological outcomes. (**a**) Serum expression levels of miR-6511b-5p. (**b**) Serum expression levels of miR-125b-1-3p. (**c**) Serum expression levels of miR-122-5p. (**d**) Serum expression levels of miR-124-3p. Horizontal lines represent the median and error bars indicate the interquartile range. * *p* < 0.05.

**Figure 2 diagnostics-11-01905-f002:**
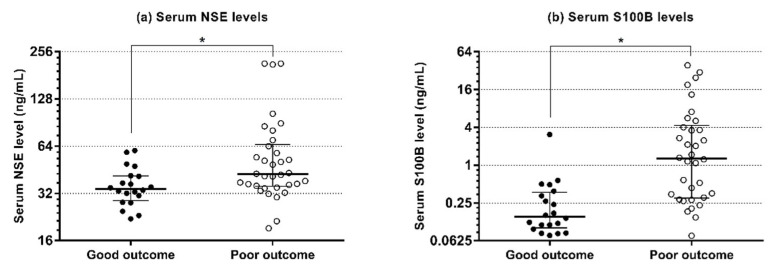
Serum levels of NSE and S100B at 6 h after return of spontaneous circulation in patients with good and poor neurological outcomes. (**a**) Serum levels of NSE. (**b**) Serum levels of S100B. Horizontal lines represent the median and error bars indicate the interquartile range. NSE—neuron-specific enolase; S100B—S100 calcium-binding protein B. * *p* < 0.05.

**Figure 3 diagnostics-11-01905-f003:**
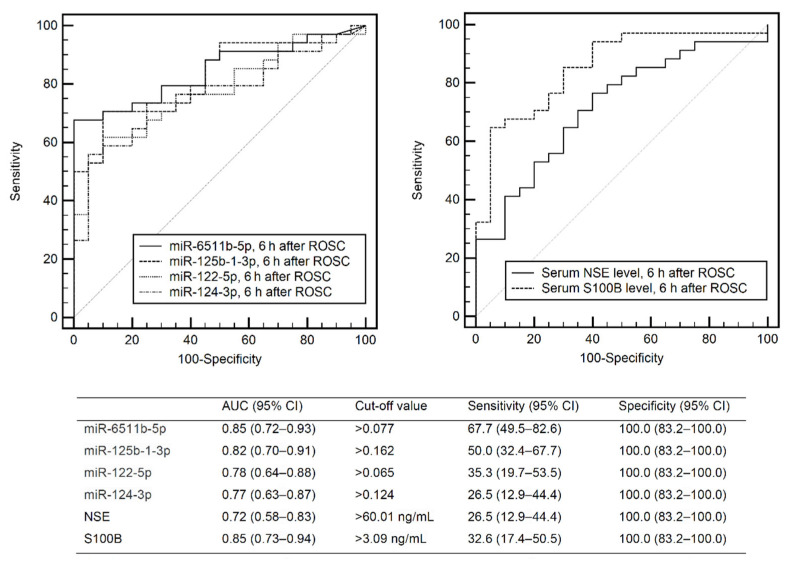
Receiver operating characteristic curves for 6-month poor neurological outcome based on the expression levels of miRNAs and NSE or S100B. NSE—neuron-specific enolase; S100B—S100 calcium-binding protein B; ROSC—return of spontaneous circulation; AUC—area under the curve; CI—confidence interval.

**Figure 4 diagnostics-11-01905-f004:**
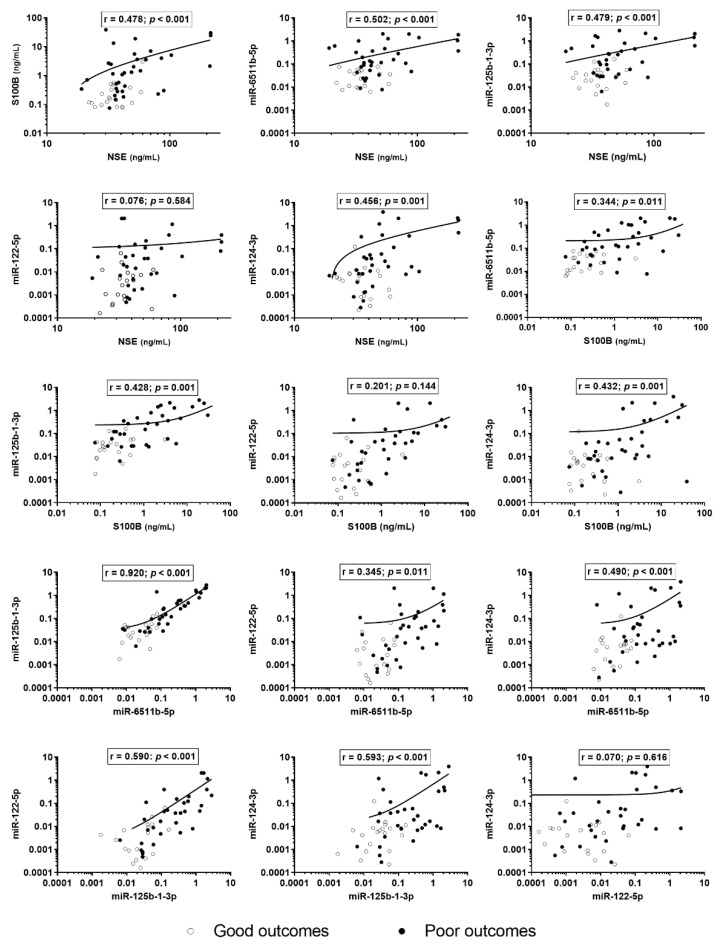
The correlation between protein and miRNA biomarkers. Log-transformed values of biomarkers and miRNA expressions are represented. Since the axis is logarithmic, values equal to zero or lower than the lowest marked value on the axis are not included on the graph. NSE—neuron-specific enolase; S100B—S100 calcium-binding protein B.

**Table 1 diagnostics-11-01905-t001:** Characteristics of the included patients.

	Good Outcome (*n* = 20)	Poor Outcome (*n* = 34)	*p*-Value
Male	12 (60.0)	25 (73.5)	0.369
Age, year	48.0 (34.0–65.58)	60.0 (44.8–72.0)	0.142
Comorbidities			
Hypertension	5 (25.0)	12 (35.3))	0.549
Diabetes mellitus	3 (15.0)	13 (38.2)	0.122
Ischemic heart disease	2 (10.0)	4 (11.8)	1.000
Chronic heart failure	1 (5.0)	1 (2.9)	1.000
Stroke	0 (0.0)	1 (2.9)	1.000
Chronic obstructive pulmonary disease	0 (0.0)	2 (5.9)	0.525
Chronic renal disease	1 (5.0)	5 (14.7)	0.395
Liver failure	0 (0.0)	1 (2.9)	1.000
Malignancy	2 (10.0)	1 (2.9)	0.548
OHCA	15 (75.0)	32 (94.1)	0.087
Cardiac cause	16 (80.0)	13 (38.2)	0.004
Shockable rhythm	13 (65.0))	4 (11.8)	<0.001
Witnessed	16 (80.0)	21 (61.8)	0.229
Bystander CPR	15 (75.0))	18 (52.9)	0.151
Time from arrest to ROSC, min	13.0 (10.0–28.8)	34.5 (16.0–46.8)	0.002
Shock on admission	2 (10.0)	17 (50.0)	0.003
Laboratory measurements			
Troponin T, ng/mL	0.1 (0.01–0.34)	0.06 (0.01–0.20)	0.706
NT-proBNP, pmol/L	10.3 (5.2–58.3)	24.2 (7.0–251.2)	0.144
Bilirubin, μmol/L	11.1 (7.4–13.7)	8.6 (5.1–12.0)	0.065
Coronary angiography	14 (70.0)	11 (32.4)	0.011
Percutaneous coronary intervention	7 (35.0)	8 (23.5)	0.530
Target temperature, 33 °C	17 (85.0)	32 (94.1)	0.347
Need for vasopressor at 6 h	7 (35.0)	24 (70.6)	0.021

Data are presented as *n* (%) for categorical variables and as the median with interquartile range (IQR) for continuous variables. OHCA, out-of-hospital cardiac arrest; CPR, cardiopulmonary resuscitation; ROSC, return of spontaneous circulation; NT-proBNP, N-terminal pro-brain natriuretic peptide.

**Table 2 diagnostics-11-01905-t002:** Prognostic performances of the different models predicting 6-month poor neurological outcome.

	AUC (95% CI)	*p*-Value
NSE	0.72 (0.58–0.83)	N/A
NSE + miR-6511b-5p	0.87 (0.78–0.96)	0.029
NSE + miR-125b-1-3p	0.86 (0.76–0.96)	0.027
NSE + miR-122-5p	0.81 (0.70–0.92)	0.121
NSE + miR-124-3p	0.75 (0.62–0.88)	0.478
S100B	0.85 (0.73–0.94)	N/A
S100B + miR-6511b-5p	0.91 (0.83–0.98)	0.281
S100B + miR-125b-1-3p	0.89 (0.80–0.97)	0.448
S100B + miR-122-5p	0.86 (0.76–0.96)	0.867
S100B + miR-124-3p	0.85 (0.74–0.95)	0.742
NSE + S100B	0.84 (0.74–0.95)	N/A
NSE + S100B + miR-6511b-5p	0.92 (0.85–0.99)	0.055
NSE + S100B + miR-125b-1-3p	0.91 (0.83–0.98)	0.074
NSE + S100B + miR-122-5p	0.86 (0.76–0.96)	0.455
NSE + S100B + miR-124-3p	0.85 (0.75–0.95)	0.629

*p*-value indicates difference between protein biomarkers and the combination model with miRNA added. AUC—area under the curve; CI—confidence interval; N/A—not applicable; NSE—neuron-specific enolase; S100B—S100 calcium-binding protein B.

## Data Availability

The data presented in this study are available on request from the corresponding author. The data are not publicly available due to legal restrictions.

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
