# Peer review of "The Levels of Circulating MicroRNAs at 6-Hour Cardiac Arrest Can Predict 6-Month Poor Neurological Outcome"

_diagnostics, 2021, doi:10.3390/diagnostics11101905_

Round 1

Reviewer 1 Report

Remove "Figure 4" from heading 3.3.

How were data tested for normal distribution?

Please give demographic data on healthy controls in comparison with CA patients (discovery phase).

Author Response

We appreciate your fine criticisms and helpful suggestions for our manuscript. We have reviewed your comments carefully, responded to the comments in a point-by-point manner, and revised our manuscript accordingly.

1. Remove "Figure 4" from heading 3.3.

We have removed “Figure 4” from heading 3.3. according to your suggestion.

2. How were data tested for normal distribution?

We have done Shapiro-Wilk test as well as Kolmogorov-Smirnov test. Distributions of 4 miRNAs were not normal.

We have added the following text to the "2.6. Statistical analysis" section

--> After testing for normal distribution, continuous variables were compared using Mann–Whitney U tests.

3. Please give demographic data on healthy controls in comparison with CA patients (discovery phase).

We have added the data on discovery cohort as supplementary Table 1.

Reviewer 2 Report

In this interesting paper, the authors investigated the potential role of circulating miRNAs as predictors of 6-h poor neurological outcome in cardiac arrest patients. Following a RNAseq-based discovery phase, the authors validated a selection of significantly altered miRNAs and correlated them with specific neurogical biomarkers. The data analysis proved a set of 4 miRNAs as independent and combined predictors of poor neurogical outcome.

The study is well done, with an appropriate experimental design and methods, extensive statistical analysis and point data discussion, including the identified limitations. I would suggest making some minor corrections:

  1. The Abstract is well structured, but full with a lot of data and I would suggest making it more easier to read by reducing these numbers and pointing the relevant results.
  2. The number of patients selected for the validation phase is missing (line 78); this is also mentioned later at line 162.
  3. I would suggest adding the levels for the evaluation of correlation coefficients (low, moderate, strong) at line 148.
  4. The signs for statistical differences between the groups for data presented in Figs 1 and 2 are missing (i.e. * sign for p<0.05).

Author Response

We appreciate your fine criticisms and helpful suggestions for our manuscript. We have reviewed your comments carefully, responded to the comments in a point-by-point manner, and revised our manuscript accordingly.

1. The Abstract is well structured, but full with a lot of data and I would suggest making it more easier to read by reducing these numbers and pointing the relevant results.

We absolutely agree. In revised abstract, we removed sentence on AUC values of combination models. We believe that it have enhanced the quality of the manuscript.

2. The number of patients selected for the validation phase is missing (line 78); this is also mentioned later at line 162.

We added the number of patients selected for the validation phase in methods section, as follows.

-->  For the validation phase, we measured the expression of candidate miRNAs and the serum levels of NSE and S100B at 6 h after ROSC in consecutive TTM-treated patients from September 2016 to July 2018 (n=54).

3. I would suggest adding the levels for the evaluation of correlation coefficients (low, moderate, strong) at line 148.

In revised manuscript, we added the levels for the evaluation of correlation coefficients as follows

--> The Pearson correlation coefficients between biomarkers was calculated (low, 0.10 to 0.29; moderate, 0.30 to 0.69; strong: 0.70 to 1.00).

4. The signs for statistical differences between the groups for data presented in Figs 1 and 2 are missing (i.e. * sign for p<0.05).

We added the asterisk signs for statistical differences in Fig 1 and 2.